# Biomimetic Construction of Artificial Selenoenzymes

**DOI:** 10.3390/biomimetics8010054

**Published:** 2023-01-28

**Authors:** Hanqing Zhao, Chengchen Xu, Tingting Wang, Junqiu Liu

**Affiliations:** Key Laboratory of Organosilicon Chemistry and Material Technology, Ministry of Education, College of Material, Chemistry and Chemical Engineering, Hangzhou Normal University, Hangzhou 311121, China

**Keywords:** selenoenzyme, glutathione peroxidase, artificial enzymes, enzyme mimic, self-assembly

## Abstract

Selenium exists in the form of selenocysteines in selenoproteins and plays a pivotal role in the catalytic process of the antioxidative enzymes. In order to study the structural and functional properties of selenium in selenoproteins, explore the significance of the role of selenium in the fields of biology and chemistry, scientists conducted a series of artificial simulations on selenoproteins. In this review, we sum up the progress and developed strategies in the construction of artificial selenoenzyme. Using different mechanisms from different catalytic angles, selenium-containing catalytic antibodies, semi-synthetic selenonezyme, and the selenium-containing molecularly imprinted enzymes have been constructed. A variety of synthetic selenoenzyme models have been designed and constructed by selecting host molecules such as cyclodextrins, dendrimers, and hyperbranched polymers as the main scaffolds. Then, a variety of selenoprotein assemblies as well as cascade antioxidant nanoenzymes were built by using electrostatic interaction, metal coordination, and host–guest interaction. The unique redox properties of selenoenzyme glutathione peroxidase (GPx) can be reproduced.

## 1. Introduction

Trace element selenium plays a protective role in the human body in the form of selenoprotein. The selenocysteine (SeC) contained in selenoproteins is the active site of selenoproteins, which makes selenocysteine have redox enzyme activity, thus providing the possibility for the realization of subsequent physiological and biochemical functions. Selenoprotein, as a mediator, mediates a series of protective effects triggered by selenium, exerting its excellent antioxidant activity in the form of various selenoproteins.

We know that natural enzymes are highly efficient biocatalysts in organisms. The orderly progress of a series of biochemical reactions in organisms depends on their precise substrate-specific recognition sites and high catalytic efficiency. During the long process of exploration, scientists have used the excellent catalytic capabilities of natural enzymes to develop biomedical medicine and materials, and also produced industrial catalysts. However, because of the inherent disadvantages of natural enzymes, such as easy loss of activity due to molecular denaturation or subunit depolymerization, as well as the limited source of natural enzymes, difficulty to purify, store and recover, etc., natural enzymes are limited in many practical applications. In order to expand the application of enzymes with actual needs, scientists have tried to develop artificial enzymes with more stable structures and lower costs as substitutes for natural enzymes in the past fifty years.

Enzyme imitation is the cornerstone of constructing biomimetic enzyme drugs and materials and obtaining high-efficiency artificial enzymes has become one of the main bottlenecks for the development of biomimetic enzyme drugs and materials. By analyzing and simulating the structure of selenoproteins, a series of artificial simulations have been carried out. In this process, scientists can have a clearer understanding of the structure and function of selenoproteins and their mechanism of action and contribute to the development of selenoprotein-like drugs and materials. In recent years, with the crossover and integration of related disciplines such as chemistry, biology, materials science, and nanoscience, the research on artificial selenoenzymes has entered a new stage of development. The research progress is rapid and many innovative research fields have emerged, such as selenium-containing antibody enzymes, molecularly imprinted selnoenzymes, artificial small molecular selenoenzymes, supramolecular selenoenzymes, semi-synthetic selenoenzymes, genetic engineered selenzymes, etc. The research scope of artificial selenoenzyme is expanding constantly, and its catalytic efficiency is also gradually improving [1,2,3,4,5,6,7,8,9]. In this review, we summed up the progress of artificial selenoenzyme glutathione peroxidase (GPx) and several innovative strategies for constructing artificial selenoenzymes.

## 2. Natural Selenoenzymes 

We know oxygen is an essential substance for organisms, but at the same time, it is also a double-edged sword for organisms. On the one hand, oxygen, as the electron acceptor at the end of the respiratory chain, combines with the paired hydrogen atoms removed from the metabolites during the respiratory chain to form water, and can continuously generate ATP (adenosine triphosphate, which provides direct energy for the body) to maintain the various metabolic activities of living organisms; on the other hand, during the process of mitochondrial oxygen consumption, active oxygen free radicals that damage themselves are generated. This is because there are some substances at the substrate end of the respiratory chain, the energy barrier is extremely low, and the two-step two-electron reduction cannot be normally performed to produce water. Usually, the one-electron reduction of O_2_ is directly performed to form superoxide anion radicals (O_2_.−). Oxygen undergoes this series of catalytic reactions to generate oxygen free radicals that are harmful to cells and can damage cells, accelerate cell aging and apoptosis, and increase the risk of diseases. Nevertheless, selenoproteins and selenoenzymes play a key role in regulating the redox environment of living life. There are many types of natural selenoproteins known, but not all selenoproteins have known functions. The most famous type of selenoproteins with known functions found in mammalian cells include GPx, deiodinases, and thioredoxin reductases. There are also some selenoproteins with definite functions, such as selenoprotein M, selenoprotein 15, selenoprotein P, selenoprotein W, and selenoprotein T, etc.

As the most representative selneoprotein, GPx is the first selenium-containing enzyme found in mammals. Moreover, its redox properties play a very important role in the whole antioxidant protection mechanism of organisms. After more than half a century of research, scientists have basically mastered the structural characteristics and functions of GPx. This enzyme possesses a very powerful antioxidant function, it can catalyze reduced hydroperoxides by glutathione. In the catalytic process, it reduces some harmful peroxides and free radicals in the body to generate water or hydroxyl compounds, thus avoiding the accumulation of peroxides and free radicals inside and outside the cells of the organism, the structure and function of the cell membrane have been protected, and the possibility of oxidative damage is reduced. The main physiological functions of GPx contains: (1) it can protect mitochondria and red blood cells from oxidative damage caused by excessive reactive oxygen species; (2) participating in the process of arachidonic acid metabolism can reduce the accumulation of superoxide lipid free radicals, avoid the occurrence of lipid peroxidation, and have the ability to antagonize the occurrence of cardiovascular diseases; (3) it has also been found to be an important part in the development of sperm. Based on the above functions, GPx has a very good effect on the treatment of various diseases such as cardiovascular and cerebrovascular diseases, Keshan disease, cataracts, energy deficiency malnutrition, and diabetes caused by excessively high levels of active oxygen in the body. It has great potential to be developed as a therapeutic drug.

Like other natural enzymes, GPx also evolved a specific binding pocket for a specific catalytic process. It has been found that the selenocysteine (SeC) residue functions as a catalytic center in GPx, and the adjacent tryptophan and glutamine form a catalytic triplet with selenocysteine. When the active center SeC participates in the catalytic reaction and binds to the substrate GSH, Arg40, Gln130, Trp148, and Arg167 form multiple hydrogen bonds and salt bridges with the carboxyl and amino groups on the GSH molecule (Figure 1) [10]. 

By the combination of multi-party forces, a meticulous catalytic network structure is formed, and a series of specific recognition and efficient catalytic physiological activities are carried out in the substrate binding groove. The catalytic mechanism and kinetics of GPx are revealed in detail, and the catalytic cycle process of GPx has been shown in Figure 2. During the entire catalytic cycle, selenol in the reduced SeC residue is first oxidized by hydrogen peroxide to generate selenic acid, which is oxidized by the affinity of glutathione (GSH), and further converts to selenyl sulfide by nuclear attack. The reaction of selenyl sulfide with a second equivalent GSH regenerates reduced selenol. Selenic acid is first oxidized by hydroperoxide to generate selenol, which is further converted into selenosulfide under the nucleophilic attack of GSH. Selenite sulfide undergoes a reduction reaction with the second equivalent of GSH, regenerates into selenic acid, and then selenic acid enters the next cycle (Figure 2**),** the entire catalytic reaction process is continuously carried out. In this process, the harmful peroxides in the organism are consumed, the content of active oxygen free radicals is reduced, and the cells are protected from oxidative damage, which is beneficial to the metabolic balance in the organism [11,12,13,14,15,16,17,18,19,20].

## 3. Artificial Selenoenzymes

### 3.1. Strategies for Constructing Artificial Selenoenzymes

#### 3.1.1. Molecularly Imprinted Selenoenzymes

Molecular imprinting technology is an efficient tool to generate specific substrate binding in artificial enzyme design. Using specific compounds as templates, these molecularly imprinted polymers can be prepared by immobilizing pre-set binding sites [21]. Through molecular imprinting technology, the imprinted cavity can be made to fit the template very well in terms of volume, shape, and functional groups. This provides an excellent idea for artificial enzyme simulation. Molecular imprinting technology can be used to prepare a binding cavity that can match the template well, and then remove the template molecule from the entire polymer system to get an imprinted cavity that is complementary in shape and size to the template molecule [22]. After obtaining the specific recognition site, active catalytic groups can be modified to the cavity to realize the simulation of an enzyme.

Currently, there are two main strategies for designing molecularly imprinted selenoenzymes. One is to design selenoenzymes with chemically synthesized polymers as the main scaffold of imprinting. It is known that one of the important ways to improve the catalytic efficiency of enzymes is to increase the recognition ability of enzymes and substrates. The binding level of catalytic centers and the direction of catalytic sites affect the catalytic activity of the enzyme. Based on the above ideas, Liu’s group reported the preparation of molecularly imprinted polymers, and a variety of molecularly imprinted polymer nanoparticles have been developed to construct artificial selenoenzymes. Based on the catalytic triplet of the GPx active center, they designed a series of arginine and selenium-containing derivatives and constructed an artificial selenoenzyme with high catalytic activity and good water solubility by using the interaction of the amino acid residues originally existing in the polymer network structure [23]. In the preparation process of molecularly imprinted polymer nanoparticles, not only the catalytic recognition ability of arginine derivatives was utilized, but also highly catalytic active tellurium was introduced on the surface of polystyrene nanoparticles so that the catalytic active tellurium center also yielded artificial enzymes with high catalytic activity [24]. 

The other strategy for molecular imprinting technology is from a bioimprinted perspective. Liu’s group tried to use protein as the main scaffold for imprinting to construct artificial selenoenzymes. First, subtilisin was used as the original scaffold, then selenium was modified on the scaffold by chemical modification, and then the bioimprinted selenium-containing enzyme was constructed by molecular imprinting technology. By directing the catalytic active center into the imprinted binding site, the constructed artificial enzyme displays significant GPx enzymatic activity, and its catalytic mechanism is consistent with the ping-pong mechanism of natural selenium enzymes (Figure 3) [25]. 

#### 3.1.2. Antibody with Selenoenzyme Activity

Another strategy to generate recognition ability and substrate binding ability is to use the concept of abzyme. Abzyme is a kind of antibody with catalytic activity prepared by biological or chemical means [26]. 

Pauling first tried to use the transition state theory to clarify the essence of enzyme catalysis. He believed that the fundamental reason for the enzyme catalysis process was that it can specifically bind and stabilize the transition state in the reaction process, thereby reducing the energy level of the reaction [27]. Therefore, a new idea for the design of artificial enzymes has been triggered. If the antibody can bind to the transition state in the reaction, it can be successfully transformed into a new enzyme with catalytic activity. Hence, scientists believe that if transition state analogs are used as antigens or haptens to immunize animals, the resulting antibodies may have a catalytic effect similar to enzymes. This is the mechanism of designing and constructing abzymes, but this method has obvious defects. If the structure of the transition state molecule is not clear, or the existing form of the transition state is relatively complicated or not analyzed clearly, then it is impossible to obtain a well-defined artificial antibody enzyme model. In order to solve this defect, scientists have tried to use a variety of methods to generate an efficient antibody enzyme. For example, using substrate analogs with hydrophobic modification as haptens, the substrate recognition sites can also be constructed, so that although the transition state of a reaction is not clear, it is also possible to obtain the active catalytic center similar to natural enzymes. If the spatial arrangement of the introduced catalytic groups and the substrate binding site is suitable, highly efficient abzymes can be produced. There are two main strategies for introducing the catalytic groups into antibody binding sites. One is the selective chemical modification method. The other is the genetic engineering site-directed mutagenesis method. Using the substrate analogs as haptens, a series of catalytic antibodies with GPx activity were prepared on the basis of standard monoclonal antibody technology. First, the GPx substrate GSH has been modified as haptens by linking the hydrophobic moieties to the two carboxyl groups of GSH, through the monoclonal antibody technology, the antibodies with GSH binding site have been generated. Then NaHSe was used to quantitatively introduce the catalytic group selenium into the hydrophobic cavity through selective selenization of active serine in antibody binding sites, thus the monoclonal antibody that consists of both a hydrophobic substrate binding site and a catalytic selenium site is prepared by monoclonal antibody technology [28]. The activity of the selenium-containing abzyme was found to be higher than that of natural GPx.

The second method for hapten design is the modification of the sulfhydryl group of glutathione by hydrophobic moieties. First, the sulfhydryl group of glutathione is chemically modified by a specific reaction between the dinitrochlorobenzene (DNCB) and sulfhydryl group. After the hapten was obtained, the hapten was cross-linked to the bovine serum albumin by using the cross-linking agent glutaraldehyde to obtain the whole antigen. Finally, the monoclonal antibody with glutathione binding site was obtained by using the preparation method of monoclonal antibody. By treating this monoclonal antibody with phenylmethylsulfonyl fluoride (PMSF) and NaHSe, the serine at the binding site of the antibody was mutated to catalytic selenocysteine. Through the above series of operations, the preparation of the monoclonal antibody with GPx enzyme activity was realized. Relevant data showed that the activity of the abzyme reached the same order of magnitude as the natural GPx activity, and its catalytic mechanism was verified to be consistent with the ping-pong mechanism of GPx [29]. 

#### 3.1.3. Cyclodextrin as Host of Artificial Selenonzyme

The construction of artificial enzymes is inseparable from the simulation structure of the enzyme binding site, so scientists aimed at some macrocycle compounds such as cyclodextrin as host molecules to mimic enzymes. Cyclodextrin is a kind of cyclic oligosaccharide because it has a slightly conical hollow cylindrical three-dimensional ring structure, and its outer shell is hydrophilic, while the inner cavity is shielded by the C-H bond, forming a hydrophobic region. This special structural feature is similar to the binding site between the enzyme and the substrate. It can provide a hydrophobic binding site like an enzyme, which means that it can act as a host molecule enveloping various types of substrates as an enzyme model. Scientists have found that the dimeric cyclodextrin system can bind a number of guest molecules with an extremely high binding ability that can almost match that of high-binding antibodies. Therefore, cyclodextrin is also an ideal scaffold for the design of artificial selenoenzymes. Using electrospray mass spectrometry and tandem mass spectrometry to analyze the binding of GSH and cyclodextrin, it was found that cyclodextrin itself can combine with GSH to form an intramolecular complex, so scientists studied a series of cyclodextrin-based the host–guest selenoenzyme model [30]. The first cyclodextrin selenoenzyme model was designed by Liu’s Group. In the selenoenzyme model β-cyclodextrin (β-CD) **1**, a diselenyl group was introduced on the surface of its narrow mouth. As a typical synthesis, the hydroxy group of β-CD was selectively tosylated and followed the nucleophilic displacement of the hydroxy group by sodium hydroselenide, which then lead to artificial selenoenzyme model **1**. This selenoenzyme model showed high GPx enzyme activity, and its activity was 4.3 times higher than that of the small molecule selenium-containing drug Ebselen, this research opened a new era for designing artificial selenoenzymes by host–guest chemistry [31]. Immediately afterward, the selenoenzyme model **2** was developed. In this model, the diselenyl moiety was introduced on the surface of the wider mouth of β-CD [32]. After testing, it was found that the activity of the selenoenyzme model **2** was significantly higher than that of model **1**, it may be that the wider cavity has a more open pocket for the substrate, which was more conducive to recognition and binding [33]. Since tellurium and selenium have similar properties, and tellurium has a stronger redox ability than selenium, scientists have introduced more active tellurium into cyclodextrin to design a selenoenzyme model. Thus, ditellurium groups were introduced into the edge of the large and small cavities of β-CD, and the obtained ditellurium cyclodextrin artificial selenoenzymes **3** and **4** was much higher than that of diselenocyclodextrin artificial selenoenzyme [34]. Then, this group wanted to make further modifications on the basis of Model **1**, so they introduced two selenol groups and cyclohexylamine on the same side, they obtained the selenoenzymes models **5** and **6**. In model **5**, when the two selenol groups were exposed to the air, they can be converted into selenic acid, and this further introduction of the catalytic group made the selenoenzyme activity of model **5** much higher than that of model **1**. For the selenoenzyme model **6**, β-CD provides a hydrophobic environment for substrate binding in its cavity, and a cyclohexylamine group is incorporated into the cyclodextrin near the catalytic selenium in order to enhance substrate affinity. Natural GPx has the ability for double substrate recognition. Scientists have found that the improvement of its catalytic activity is mainly due to the fact that the binding site of the substrate is very consistent with the shape and size of the substrate, and the introduced imino group is conducive to the stability of selenate and the improvement of catalytic reactions. Another improved binding method has emerged, which can introduce the second binding site into the system. It is also common in natural enzyme systems and can promote substrate binding and transition state recognition. The study of cyclodextrin dimer selenoenzyme models **7** and **8** also proved that the strategy is feasible (Figure 4). By connecting selenium-containing molecules to cyclodextrins, we can simulate a series of different selenium-containing enzymes (Figure 4
**9**–**15**). The steady-state kinetics of the catalytic reaction were found to be consistent with the ping-pong mechanism of GPx [35,36,37,38]. 

#### 3.1.4. Dendrimers and Hyperbranched Polymers as Hosts of Artificial Selenonzymes

As mentioned above, cyclodextrin molecules as hosts can accommodate guest molecules, then dendrimers and hyperbranched polymers are also good choices as host molecules, because they have complex three-dimensional topology structures, and can also accommodate a large number of hydrophobic guest molecules [39,40,41]. The structure of dendrimers, like natural enzymes, can be controlled accurately at the molecular level [42,43,44,45,46]. Dendritic macromolecules display the ability to capture substrates during the entire reaction process. The good trapping ability makes it a favorable scaffold for artificial enzyme mimics [47,48,49,50,51].

Therefore, the scientists introduced selenium and tellurium into the center of dendrimers in order to obtain artificial selenoenzyme with high GPx activity. A classic model mentioned here is the dendritic macromolecule with the introduction of the diselenium catalytic center shown in Figure 5a, which firmly binds the substrate at the active center, making its GPx activity increase to 2431.20 uM min^−1^, which is 1400 times that of the organic selenoenzyme Ebselen [52]. It is found that the size of the dendritic scaffold has a relatively large impact on the catalytic activity. The longer the scaffold extends, the stronger its influence on the hydrophobic microenvironment, and the corresponding higher GPx enzyme activity. Similarly, as shown in Figure 5b, tellurium is also introduced into the dendrimer scaffold. In theory, its GPx enzyme activity will be higher than that of diselenide dendrimers. In fact, the observation is the same, but with a slightly special point. The enzyme activity of the tellurium-containing model decreased at first and then increased from the 0th generation to the 2nd generation. The main reason is that the periphery of ditellurium dendritic selenomimetic enzymes is composed of oligomer chains, which are much larger than the peripheral groups (Figure 5a). Then, a hyperbranched polyselenium compound with multiple catalytic active centers was constructed. The difference between this model and the dendrimer selenoenzyme is that it has a lot of catalytic active centers and provides many catalytic positions, which is conducive to greatly improving the activity of artificial selenoenzymes (Figure 5c) [53,54,55,56,57,58,59].

#### 3.1.5. Semisynthetic Selenoenzyme by Chemical Mutation

The semi-synthetic enzymes, as the name suggests, use some natural proteins with special structures as scaffolds to modify them on the basis of their inherent structures, and introduce new catalytic functions on the scaffold to realize the artificial imitation of enzymes [60]. Scaffolds for constructing semisynthetic selenoenzyme models have been widely reported, for example, the rat glutathione transferase (GST) T2-2 class (rGST T2-2) which has the highly specific binding to glutathione has been used to prepare semisynthetic selenoenzyme. In the active site of GST, the serine residue can be transformed into selenocysteine by means of chemical modification, combined with the precise substrate binding and positioning of the catalytic group selenocysteine. The GPx activity of the obtained semisynthetic selenoenzyme shows at least 4000 times that of Ebselen, achieving a significant increase in enzyme activity, and its activity is even close to that of some natural GPx enzyme activities [61,62,63].

Later, another GST variant, human glutathione transferase zeta 1(hGSTz-1), was reported. This GST variant was different. The crystal structure of hGSTz-1 showed that its active site was located between the N-terminal and C-terminal domains. In the deep crevice, there were three very highly conserved residues (Ser14–Ser15–Cys16), and Ser 14 and Ser 15 are closed to the thiol of GSH that binds to the active site. After activation by PMSF, then Ser can be easily chemically mutated to SeC using NaHSe as a nucleophile [64,65]. It was found that the GPx activity of se-hGSTz-1 was even higher than that of rabbit liver [66]. It can be seen that the accuracy of the binding site and the orientation of selenium in the binding site have a direct correlation with the activity of GPx [67]. These studies have brought us new thinking that we can precisely introduce catalytic active groups at naturally evolved binding sites to realize the highly efficient design of semi-synthetic selenoenzyme.

#### 3.1.6. Selenoenzyme Design by Genetic Engineering

The catalytic activity of GPx enzymes mainly comes from the selenocysteine on the catalytic site. When using gene engineering technology to simulate selenoenzyme, the important problem encountered is how to introduce the target catalytic amino acid selenocysteine (SeCys). As the 21st amino acid, selenocysteine requires special conditions to embed in proteins. It is encoded by the base UGA, which is also the sequence of the stop codon, so it is difficult to express directly through genetic recombination technology. An alternative way to introduce selenium into protein structure in a biological way is to use an auxotrophic expression system. Cowie and Cohen were the first to have this idea and put it into practice [68,69,70,71]. They used the E. coli expression system to introduce selenium into the active center of the protein by using its own metabolism and realized the biosynthesis of selenium for the first time [72]. Then, researchers began to widely use this auxotrophic expression system for protein transformation, and the cysteine of the protein was substituted into selenocysteine. Earlier, Liu’s group used a semi-synthetic method to convert the active site serine of GST into selenocysteine, and then prepare a semi-synthetic selenoenzyme by chemical modification. Although this method can achieve selenoenzyme with higher GPx activity, the chemical modification method has certain limitations. It has no way to accurately target the amino acid residues in the active center, resulting in uncertainty in the process of selenization. Thus, there is no way to accurately simulate the catalytic pocket of selenoenzyme, which is not conducive to subsequent studies on the structure and function of selenoenzymes. In contrast, genetic engineering is a very precise and site-selective manner. Liu’s group introduced selenocysteine into the binding site of GST through genetic engineering method, mutated the serine at position 9 to cysteine, and then used auxotrophic expression to convert the cysteine at position 9 to selenocysteine, which successfully achieved the incorporation of selenium [73]. After testing, it was found that the expressed protein displayed high GPx activity, and the activity reached the same order of magnitude as the natural GPx activity, which proved the feasibility of this method.

Based on this, Liu and coworkers began to study other enzyme systems which have similar GSH binding sites to GPx. Glutaredoxin (Grx) belongs to the thioredoxin folding family, has the same Trx folding domain, and possesses the ability to specifically recognize the substrate glutathione. They first cloned the functional Grx domain from mouse thioredoxin glutathione reductase (Tgr), then carried out site-directed mutation on Grx to transform the Cys48 in Grx active site into SeC, and finally realized the high-efficiency expression selenoGrx. GPx activity of selenoGrx was also comparable to that of natural GPx. Steady-state kinetic studies showed that its catalytic mechanism also conformed to the ping-pong mechanism [74,75,76,77,78,79].

Next, scientists were not satisfied with only transforming enzymes with similar GSH binding sites to GPx into artificial selenoenzyme, they tried to find another protein with the potential to be transformed into GPx, thus achieving the goal of transforming ordinary proteins into artificial selenoenzyme. Then, they turned their attention to the stable protein one (SP1) system. The SP1 protein is a highly stable ring-shaped protein composed of twelve identical monomers, which provides a premise scaffold for the construction of selenium-mimetic enzymes. Subsequently, they introduced selenocysteine, the active center of glutathione peroxidase, on the surface of the SP1 protein by means of computational design and genetic engineering, and successively selected positions 57, 84, and 98, and realized the construction of selenium-containing SP1 with high GPx activity. This artificial selenoenzyme has excellent application prospects due to two main characteristics. Firstly, because of the dodecamer structure of the SP1 protein, twelve active selenocysteine sites were incorporated, making its high catalytic activity; secondly, due to the inherent high-temperature resistance of the SP1 protein, the constructed artificial selenoenzyme showed a high degree of thermostability, which made the artificial selenoenzyme have a broader application prospect [80,81,82,83,84].

### 3.2. Multiple Assembly of Artificial Selenoenzymes

After constructing a variety of selenoenzyme enzymes, researchers try to assemble the constructed selenoenzymes to form a variety of assemblies with regular morphology. The selenoprotein assemblies would endow high stability and different perspectives for medicine and material application. 

#### 3.2.1. Assembly of Artificial Selenoenzymes by Electrostatic Interaction

As the driving force of supramolecular assembly, the electrostatic interaction is one of the important means to construct supramolecular assemblies. Through the computational simulation, it was found that the surface of the SP1 protein was negatively charged and distributed symmetrically. Based on the multiple electrostatic interactions, Liu’s group designed a variety of supramolecular nano-assemblies. One example is to introduce CdTe quantum dots with opposite charges on the surface to interact with negatively charged SP1 protein, and through electrostatic interactions, highly ordered protein nanowires with sandwich structures have been constructed (Figure 6a) [85].

Another method has been reported by using two positively charged ethylenediamine molecules as linkers. First, the active center SeC is introduced at position 57 of SP1 by genetic engineering and auxotrophic expression technology, and then the surface band is induced by electrostatic interaction. Negatively charged SP1 can self-assemble with positively charged ethylenediamine molecules, and a selenoprotein nanotube with GPx function has been developed (Figure 6b) [86]. In a similar way, the selenated SP1 protein was assembled and dissociated by the interaction of the positively charged dendrimer PAMAM with selenated SP1 protein by adjusting the ionic strength and changing the electrostatic interaction between them (Figure 6c) [87]. It was worth noting that selenocysteine was introduced at position 6, so the active center was located in the inner center of the SP1 protein ring. When the proteins were assembled with the electrostatic force, the catalytic center was blocked by the small dendrimers, and the substrate cannot enter into the catalytic pocket, so the GPx activity was lost. However, when the ionic strength was adjusted again and the electrostatic force changed, resulting in the disassembly of the protein assemblies, the catalytic central part was exposed again, and the activity of the selenoenzyme was restored. The selenized SP1 protein showed an activity of 117.2 umol·min^−1^·umol^−1^, while the SP1 protein without SeC center had no obvious GPx activity. At the same time, the selenoenzyme activity was switched by tuning the electrostatic forces, which provided a basis for future catalysis in vivo for recyclable catalytic reactions.

#### 3.2.2. Assembly of Artificial Selenoenzymes by Metal Chelation Force-Coordination

The metal chelation force has the characteristics of short time and high efficiency, so if the metal ion chelation force is used to assemble, this should be an alternative way of protein assembly. As a model protein GST from Schistosoma japonicum exhibits a natural dimerization in which its two subunits are non-covalently bound and are linked via the middle biaxial connection, this specific symmetry makes it possible to become the main scaffold of the protein assembly. In the beginning, a gene-histidine tag was introduced into the middle core structure of the GST dimer protein, then the histidine tag stretched out from the middle of the GST to expose it to nickel ions, and linkage was achieved through the force of metal ion coordination. Since the fused metal histidine tag was located in the middle of the GST dimer, protein nanowires with good enzymatic activity were finally formed [88,89]. Subsequently, getting more inspiration from this work, precise control of the GST assembly protein has been carried out by changing the position of histidine. It is known that when the metal coordination binding site is on the C2 axis, it is conducive to the formation of protein nanowires. When designed two binding sites are designed to be located on the GST protein shoulders, they show a “v”-shaped angle, and perpendicular to the C2 axis of the GST protein, the curved extension of the protein assembly can be achieved. The introduction of two adjacent histidine residues is meaningful, as the formation of a four-coordinated nickel ion chelation site requires two histidines at one position. After calculation and simulation, it was found that if the Cys at position 137 of GST is mutated into His, it can form a double histidine clamp with the original amino acid His at position 137 adjacent to it, which is conducive to the realization of efficient metal coordination. Additionally, in this mode, a structural non-covalent interaction network can be formed between two proteins. Through the electrostatic interaction inside the network and the combined effect of hydrogen bonding, finally, GST-based protein nanorings have been developed successfully [90]. Based on this assembly method, the sixth or ninth site of the GST protein is mutated into cysteine, and then the auxotrophic expression system is used to selenize the GST in order to achieve GPx activity. The metal coordination force has been proven to be a powerful strategy to realize the construction of artificial selenoenzyme on nanostructures.

#### 3.2.3. Assembly of Artificial Selenoenzymes by Host–Guest Interaction 

GST, which evolves the glutathione-binding cavity, has been demonstrated to be a good selenoenzyme building block for constructing assembled nanostructures [91]. Liu’s group attempted to construct supramolecular selenoprotein assemblies based on non-covalent interactions. They chose position 6 of the GST protein to be mutated and transformed into cysteine, which is conducive to the auxotrophic expression and the incorporation of selenium so that cysteine can be converted into selenocysteine. On this basis, a short peptide tag composed of three amino acids FGG is fused to the N-terminus of the GST protein. This FGG short peptide tag has been proven to have multiple hydrogen bonds, ion-dipole interactions, and hydrophobicity with the host molecule cucurbituril CB [8] (Figure 7a) [92]. By combining the π–π interaction and other forces, the specific supramolecular binding constant of CB [8] with FGG is as high as 10^9^–10^11^M^−2^. The two N-terminals conformational positions of the GST protein dimer are opposite, and a small FGG tag protrudes on two sides of seleno-GST, which can specifically combine with the middle cavity of the CB [8] molecule and is conducive to the formation of ordered selenium-containing protein assemblies. Finally, the selenized GST protein nanowires were successfully constructed. After testing, it was found that they display high-efficiency GPx activity and conform to the ping-pong mechanism similar to the GPx-catalyzed reaction process, and their GPx activity is higher than that of unassembled single selenized GST.

The construction of artificial selenoenzymes from monomers to assemblies has two major significant advantages. On the one hand, the number of active sites possessed by the selenoenzymes is greatly increased, which is not available for monomeric selenoenzymes molecules. By adjusting the external environment, the size of the assemblies can be controlled, which gives the selenoenzymes activity of the selenoenzymes assemblies adjustable and even enables the “on” and “off” of selenoenzymes activity. On the other hand, the tandem association of selenoenzymes monomer molecules was achieved by electrostatic interaction, metal chelating force coordination, host–guest interaction, and the assemblies could be found to have a highly uniform and homogeneous distribution. The highly ordered nature of the assemblies gives the constructed nanoenzymes the potential to be developed as functional bionanomaterials. The special properties of the substances incorporated as tandem agents allow the selenoenzymes assemblies to have extremely high GPx activity along with the properties of tandem agents for functional stacking and upgrading.

### 3.3. Multi-Enzyme Cascade Nanozyme Model

In organisms, most enzymes work in a coordinated manner. The various pathways and processes of metabolism often require the participation of many enzymes and then complete the reaction process in sequence, and finally realize the corresponding physiological effects that maintain the health and balance of organisms [95,96].

We know that GPx, superoxide dismutase (SOD), and catalase (CAT) in organisms are of great significance to the physiological metabolism of the body, and they together constitute the mechanism for removing reactive oxygen species and are the main defense system to protect cells from peroxidative damage. Therefore, the construction of a nano-assembly system loaded with cascade antioxidative enzymes can protect cells from oxidative damage by a cascade antioxidative means.

Artificial nano-selneoenyzme can also be constructed by the self-assembly of amphiphilic polymers. Based on the host–guest interaction between cyclodextrin and adamantane, Liu’s group synthesized tellurated cyclodextrin with GPx active center, and adamantanes modified with tetrapyridyl manganese porphyrin as SOD mimic (MnTPyP-M-Ad) have been non-covalently connected together through host–guest complexation. Thus, a bifunctional enzyme model was first designed, the synthesized block copolymer, β-CD-PEG-b-PNIPAAm, consists of temperature-responsive poly-N-isopropylacrylamide and a tellurium active center. The resulting β-CD-PEG-b-PNIPAAm-Te molecule was assembled with MnTPyP-M-Ad, thus a bifunctional artificial enzyme with both SOD and GPx activities, and a unique temperature response function has been constructed. Studies have shown that the enzyme exhibits stable GPx and SOD activities, and when the temperature is changed, both SOD and GPx activities can be tuned. The temperature-responsive properties of the artificial enzyme make the enzyme model adjustable (Figure 7b) [93].

The cascade antioxidant nanoenzyme with dual activities of SOD and GPx can also be constructed by protein self-assembly. In order to realize the construction of a dual-enzyme system, the glutathione binding site on the surface of the SP1 protein ring was designed by computer simulation, the 57th gene was mutated to the cysteine in the center of the groove, and then the catalytic active center SeC was introduced into SP1 protein by auxotrophic expression. At the same time, the dendritic molecule PAMAM was modified with manganese porphyrin, a SOD catalytic center, followed by electrostatic interaction, a protein nanowire with bifunctional enzyme activities has been prepared by self-assembly of selenized SP1 with dendrimers. The double-enzyme cascaded nanowire has demonstrated an excellent function for protecting mitochondria from oxidative stress damage, and studies have also found that this double-enzyme exhibits a more significant antioxidative damage effect than a single enzyme (Figure 7c) [94]. The design of an antioxidative nanoenzyme system with cascade catalytic performance is an important step to further develop biomedical materials and drug delivery systems in the future [97,98].

The metabolic pathways and processes of organisms require the participation of multiple enzymes, which then follow an ordered pattern to accomplish physiological activities. Based on the assembly of a single species of artificial selenoenzyme, a variety of different enzymes are introduced to act synergistically. The construction of a nano-assembled system containing a cascade of antioxidant enzymes can protect cells from oxidative damage by means of a cascade of antioxidants. This puts the simulation of artificial selenoenzyme on a more biomimetic ladder, which broadens up the physiological significance of artificial selenoenzyme.

## 4. Conclusions

After a long evolutionary journey, organisms have developed an elaborate antioxidant defense system within themselves. Various antioxidants in the organisms such as GPx, SOD, and CAT, which have played their respective roles at different molecular levels, have played a pivotal role in the stability and homeostasis of the organisms.

Scientists have proposed various strategies to design artificial selenoenzymes to address the problems of instability and lack of sources of natural selenoenzymes. Combining substrate recognition with the catalytic selenium moieties and introducing catalytic selenium through genetic engineering and chemical site-specific introduction, a series of artificial selenoenzymes with highly efficient GPx activities have been developed. For example, selenium-containing catalytic antibodies, semi-synthetic selenonezyme, and selenium-containing molecularly imprinted enzymes have been constructed by using different mechanisms from different catalytic angles. Taking advantage of host molecules such as cyclodextrin, dendrimers, and hyperbranched polymers as the main scaffolds, more synthetic selenoenzyme models have been designed. Then, a variety of selenoprotein assemblies as well as cascade antioxidant nanoenzymes were built by using electrostatic interaction, metal coordination, and host–guest interaction. 

Artificial simulation of selenoenyzme not only helps to understand the relationship between selenoenzyme structure and function, but more importantly, the antioxidant capacity of selenoenzyme can protect organism cells from oxidative stress, and further avoid the accumulation of harmful substances inside and outside the cell membrane, and finally achieve the dynamic balance of life, health, and metabolic activities of organisms. 

The dilemmas and challenges for the future artificial selenoenzyme are not only to improve its selenoenzyme activity but more importantly to give it multifunctionality. Scientists should not only simulate the structure of selenoenzyme but also simulate the environment in which the artificial selenoenzyme is located to realize the interaction between the artificial selenoenzyme and the surrounding environment, so as to lay the foundation for the future artificial selenoenzyme to really become a clinical drug. It will have extraordinary significance for the future of human health.

## Figures and Tables

**Figure 1 biomimetics-08-00054-f001:**
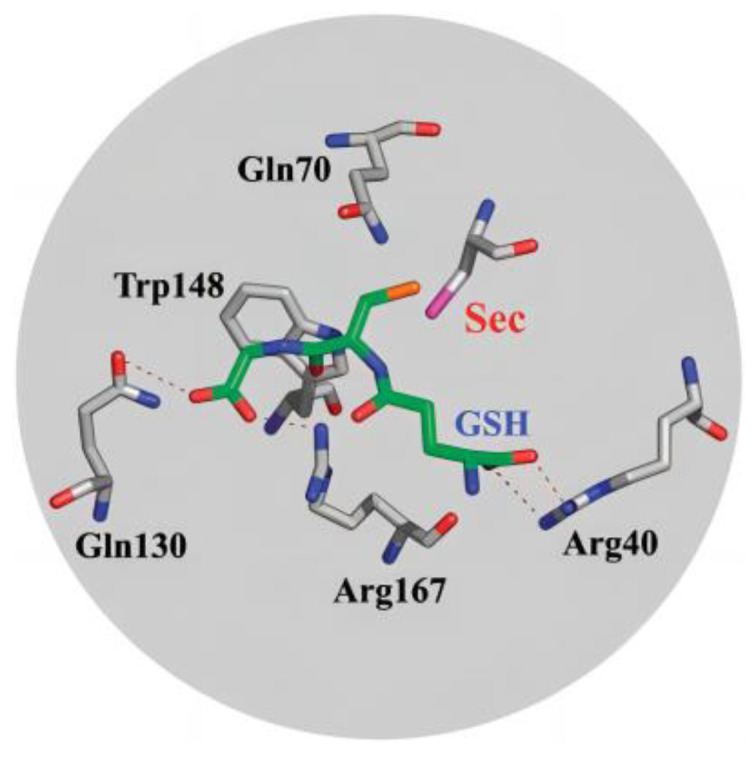
cGPx activity site map: structure for GSH and Arg40, Arg167, and Gln130. Reproduced with permission from X. Huang, et al. [10].

**Figure 2 biomimetics-08-00054-f002:**
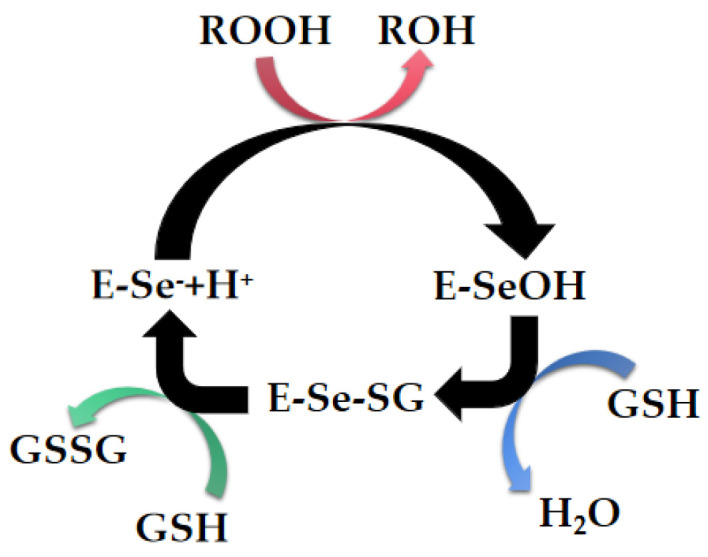
The catalytic cycle of GPx.

**Figure 3 biomimetics-08-00054-f003:**
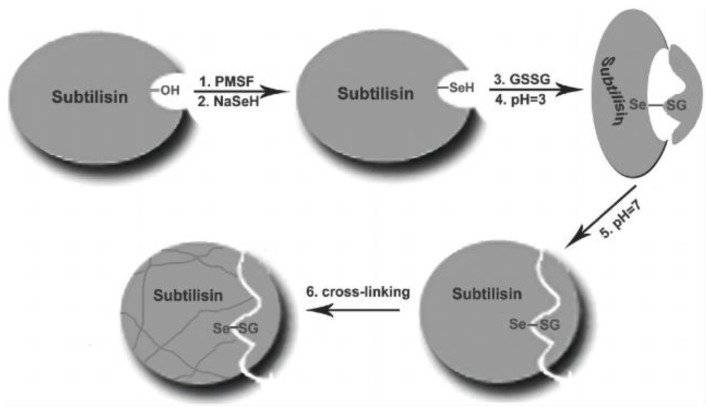
Artificial selenoenzyme designed by bioimprinted strategy. Reproduced with permission from L. Liu, et al. [25].

**Figure 4 biomimetics-08-00054-f004:**
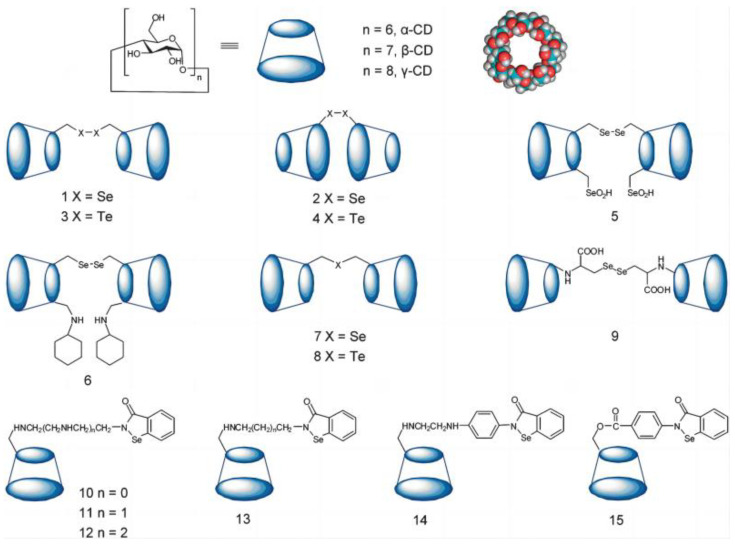
Graphic representation of a series of cyclodextrin-based selenoenzyme mimics. Reproduced with permission from X. Huang, et al. [10].

**Figure 5 biomimetics-08-00054-f005:**
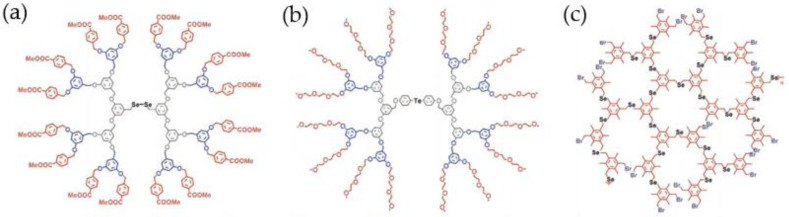
Selenoenzyme model based on dendrimers and hyperbranched polymers. (**a**) Dendrimers containing active sites of selenium. (**b**) The second-generation dendrimers with telluride moiety. (**c**) Hyperbranched compound model of selenium-containing catalyst. Reproduced with permission from X. Huang, et al. [10].

**Figure 6 biomimetics-08-00054-f006:**
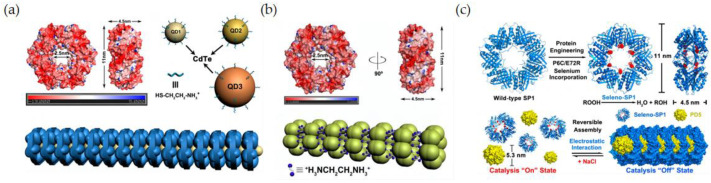
Selenium-containing enzyme mimics constructed based on assembly electrostatic interactions. (**a**) Quantum dots with SP1 protein. Reproduced with permission from L. Miao et al. [85]. (**b**) EDA with SP1 protein. Reproduced with permission from L. Miao et al. [86]. (**c**) PAMAM with SP1 protein. Reproduced with permission from T. Pan et al. [87].

**Figure 7 biomimetics-08-00054-f007:**
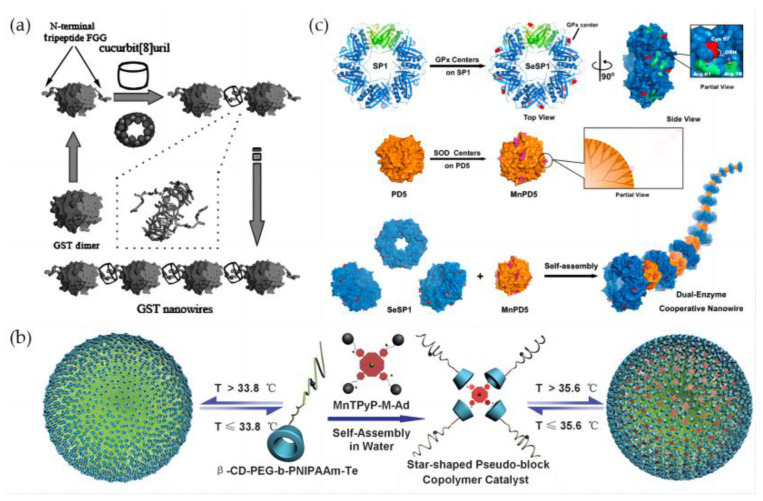
Different assembly methods of artificial selenoenzymes. (**a**) Graphic representation of selenium-containing protein nanowires formed by host–guest interaction based on CB [8] and FGG. Reproduced with permission from C. Hou et al. [92]. (**b**) Graphical representation of a bifunctional selenoenyzme model with temperature response. Reproduced with permission from S. Yu et al. [93]. (**c**) Diagram of self-assembly of circular proteins induced by “soft nanoparticles” with multi-enzyme synergistic antioxidant capacity..Reproduced with permission from H. Sun et al. [94].

## Data Availability

Not applicable.

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
