# Peer review of "Biomimetic Construction of Artificial Selenoenzymes"

_biomimetics, 2023, doi:10.3390/biomimetics8010054_

Round 1

Reviewer 1 Report

The overall content of this review is clear and complete, and the discussion and conclusions are relatively reasonable. This review gives a very detailed and complete introduction to the construction methods of artificial selenoenzyme glutathione peroxidase (GPx) in recent years, and cites many classic studies in related field, which should exhibit reference significance for development and practical medical application of artificial selenoenzyme GPx. The following minor issues should be dealt prior to be published:

1. The title of the review is “ biomimetic construction of artificial selenoenzymes ”, however in the third part (artificial selenoenzymes), many methods only involve chemical construction including molecular imprinting, host-guest, electrostatic interaction and supramolecular self-assembly, how is the biomimetic construction reflected here? What is the difference between the structure of natural GPX and artificial GPX?

2. When hGSTZ1-1 was mentioned for the first time, the full name was not given, but it was given in the next sentence. The abbreviations of phenylmethylsulfonyl fluoride and benzylsulfonyl fluoride are both PMSF, whether they are the same substance ?

3. This review classifies various artificial GPX construction strategies and follows the order from single artificial GPX to multi-enzyme assembly. However, it lacks horizontal comparison between different construction methods, and the characteristics of different methods are not well reflected.

Author Response

The overall content of this review is clear and complete, and the discussion and conclusions are relatively reasonable. This review gives a very detailed and complete introduction to the construction methods of artificial selenoenzyme glutathione peroxidase (GPx) in recent years, and cites many classic studies in related field, which should exhibit reference significance for development and practical medical application of artificial selenoenzyme GPx. The following minor issues should be dealt prior to be published.

Thank you very much for your decision and constructive comments on my manuscript. We are very grateful to Reviewer for reviewing the paper so carefully.We have tried our best to improve the manuscript and the revision notes point-to-point are given as follows:

Comment 1

The title of the review is “ biomimetic construction of artificial selenoenzymes ”, however in the third part (artificial selenoenzymes), many methods only involve chemical construction including molecular imprinting, host-guest, electrostatic interaction and supramolecular self-assembly, how is the biomimetic construction reflected here? What is the difference between the structure of natural GPX and artificial GPX?

Response 1

Thank you for bringing these meaningful questions to our attention. The bionic structure is reflected in the simulation of the active center and the catalytic binding pocket of the natural selenoenzymes. The mimicry of the constructed selenoenzymes assemblies presented in the third part lies in the mimicry of the monomer and the self-assembly of the monomeric selenoenzymes to better visualize and control the enzyme so that the constructed assemblies of selenoenzymes can be better utilized.

The most prominent difference between the structures of natural GPx and artificial GPx is that the number of sites in the active center and the number of binding pockets that artificial GPx has are different from natural GPx and can be changed. Artificial selenase can also achieve controlled self-assembly, and regular nanostructures can be formed by many different forces, while superposition of selenase activity is achieved because of the assembly.

The highly ordered nature of the assemblies gives the constructed nanoenzymes the potential to be developed as functional bionanomaterials. The special properties of the substances incorporated as tandem agents allow the selenoenzymes assemblies to have extremely high GPx activity along with the properties of tandem agents for functional stacking and upgrading.

Comment 2

When hGSTZ1-1 was mentioned for the first time, the full name was not given, but it was given in the next sentence. The abbreviations of phenylmethylsulfonyl fluoride and benzylsulfonyl fluoride are both PMSF, whether they are the same substance ?

Response 2

Thank you for reminding us these important points. We feel sorry for our carelessness. In our resubmitted manuscript, we have given the full name of hGSTz-1 when it was first mentioned (On page 8, line 336).

As for the second question, the abbreviation of phenylmethylsulfonyl fluoride is PMSF but benzylsulfonyl fluoride is not. We have carefully checked the manuscript and corrected the errors accordingly (On page 5, line 220).

Comment 3

This review classifies various artificial GPX construction strategies and follows the order from single artificial GPX to multi-enzyme assembly. However, it lacks horizontal comparison between different construction methods, and the characteristics of different methods are not well reflected.

Response 3

We sincerely thank you for bringing these meaningful points to our attention. We have added the characteristics and horizontal comparison between different construction methods in the manuscript in Page 11 lines 493-506,Page 13 lines 560-567.

Reviewer 2 Report

In this manuscript, the strategies for constructing artificial selenoenzymes and the multiple assembly of artificial selenoenzymes were summarized. Selenoproteins and selenoenzymes played an important role in regulating the redox environment of the living life. Due to the lack of sources and instability of natural selenoenzymes, many artificial selenoenzymes were developed to make up the disadvantage of natural selenoenzymes and even showed higher catalytic activity and have potential to protect organism cells from oxidative stress, and further avoid the accumulation of harmful substances. The development of artificial selenoenzymes benefit to be developed as practical drugs. This review concluded the progress and development of artificial selenoenymes well, but the manuscript need further modification before being published.

1.     There were some grammatical mistake, such as ‘development The research progress’ in line 48, ‘3.1. Strategies for constructing artificial selenoenzymes’ without capitalization in line 124, ‘GPx also evolve a specific binding pocket’ in line 96.

2.     The advantages of the ordered structure formed by artificial enzyme self-assembly over artificial enzyme monomer were missing.

3.     Challenges and dilemmas of further development of artificial biomimetic enzymes and the further research t for an artificial enzyme to develop into a drug were expounded clearly in the manuscript.

Author Response

In this manuscript, the strategies for constructing artificial selenoenzymes and the multiple assembly of artificial selenoenzymes were summarized. Selenoproteins and selenoenzymes played an important role in regulating the redox environment of the living life. Due to the lack of sources and instability of natural selenoenzymes, many artificial selenoenzymes were developed to make up the disadvantage of natural selenoenzymes and even showed higher catalytic activity and have potential to protect organism cells from oxidative stress, and further avoid the accumulation of harmful substances. The development of artificial selenoenzymes benefit to be developed as practical drugs. This review concluded the progress and development of artificial selenoenymes well, but the manuscript need further modification before being published.

Thank you very much for your decision and constructive comments on our manuscript. We are very grateful to Reviewer for reviewing the paper so carefully. We have tried our best to improve the manuscript and the revision notes point-to-point are given as follows:

Comment 1

There were some grammatical mistake, such as ‘development The research progress’ in line 48, ‘3.1. Strategies for constructing artificial selenoenzymes’ without capitalization in line 124, ‘GPx also evolve a specific binding pocket’ in line 96.

Response 1

We feel sorry for our carelessness. We will be happy to edit the text further, based on helpful comments from the reviewers. We have revised the manuscript in Page 2 line 56, Page 3 line 130, Page 3 line 105, Page 4 line 135, Page 6 line 242, Page 6 line 266.

Comment 2

The advantages of the ordered structure formed by artificial enzyme self-assembly over artificial enzyme monomer were missing.

Response 2

 Thank you for bringing these meaningful points to our attention. We have added the advantages of the ordered structure formed by artificial enzyme self-assembly over artificial enzyme monomer in the manuscript in Page 11 lines 493-506, Page 13 lines 560-567.

Comment 3

Challenges and dilemmas of further development of artificial biomimetic enzymes and the further research for an artificial enzyme to develop into a drug were expounded clearly in the manuscript.

Response 3

Thank you for bringing these meaningful points to our attention. We have revised the Conclusions in the manuscript in Page 13 lines 594-600.

Reviewer 3 Report

The manuscript of “Biomimetic Construction of Artificial Selenoenzymesis interesting to read and appreciated for good attempt. Author explained about several things like, artificial selenoenzyme glutathione 53 peroxidase (GPx) and several innovative strategies for constructing artificial selenoen- 54 zymes. I decided minor revision and also correct the following suggestion before accept the manuscript.

1.      The abstract did not fulfill all the results and it should be written corrected with obtained results.

2.      Keywords should be in order and relate to the manuscript

3.      The intro part does not have any single references. why?

4.      Some many references are old like ten years ago, try t to add new references.

5.       Conclusion should be short and communicate your manuscript

Author Response

The manuscript of “Biomimetic Construction of Artificial Selenoenzymes” is interesting to read and appreciated for good attempt. Author explained about several things like, artificial selenoenzyme glutathione 53 peroxidase (GPx) and several innovative strategies for constructing artificial selenoen-54 zymes. I decided minor revision and also correct the following suggestion before accept the manuscript.

Thank you very much for your decision and constructive comments on our manuscript. We are very grateful to Reviewer for reviewing the paper so carefully. We have tried our best to improve the manuscript and the revision notes point-to-point are given as follows:

Comment 1

The abstract did not fulfill all the results and it should be written corrected with obtained results.

Response 1

Thank you for bringing these meaningful points to our attention. We have revised the Abstract in the manuscript in Page 1 lines 14-23.

Comment 2

Keywords should be in order and relate to the manuscript.

Response 2

We sincerely thank you for your careful reading. We have revised the Keywords in the manuscript in Page 1 line 24.

Comment 3

The intro part does not have any single references. why?

Response 3

We sincerely thank you for your careful reading. The citation format for the references in the intro part of the manuscript refers to the citation format of other articles in the journal.

Comment 4

Some many references are old like ten years ago, try to add new references.

Response 4

Thank you for reminding us this important point. We have added new references in the manuscript ( References:[22], [35], [36], [37], [39], [40], [44], [48], [50], [54], [55], [59], [62], [63], [73], [76], [78] ).

Comment 5

Conclusion should be short and communicate your manuscript.

Response 5

 Thank you for bringing these meaningful points to our attention. We have revised the Conclusions in the manuscript in Page 13 lines 570-600.